# A Gap Between the Hypergraph and Stabilizer Entropy Cones

**Ning Bao,**[a,b] **Newton Cheng,**[b] **Sergio Hernández-Cuenca,**[c] **Vincent Paul Su**[b]

[a]*Computational Science Initiative, Brookhaven National Lab, Upton, NY, 11973, USA*

[b]*Center for Theoretical Physics, Department of Physics, University of California, Berkeley, CA 94720, USA*

[c]*Department of Physics, University of California, Santa Barbara, CA 93106, USA*

*E-mail:* ningbao75@gmail.com, newtoncheng@berkeley.edu, sergiohc@ucsb.edu, vipasu@berkeley.edu

ABSTRACT: It was recently found that the stabilizer and hypergraph entropy cones coincide for four parties, leading to a conjecture of their equivalence at higher party numbers. In this note, we show this conjecture to be false by proving new inequalities obeyed by all hypergraph entropy vectors that exclude particular stabilizer states on six qubits. By further leveraging this connection, we improve the characterization of stabilizer entropies and show that all linear rank inequalities at five parties, except for classical monotonicity, form facets of the stabilizer cone. Additionally, by studying minimum cuts on hypergraphs, we prove some structural properties of hypergraph representations of entanglement and generalize the notion of entanglement wedge nesting in holography.

## 1 Introduction

The entanglement structure of a general quantum state can be studied by considering the von Neumann entropies of all of its subsystems, i.e. all possible unions of parties resulting from a factorization of the Hilbert space. The quantum entropy cone captures the set of entanglement entropies of subsystems which are physically realizable by quantum states. The structure of this object for mixed states on 4 and more parties remains an open question in quantum information theory. In recent years, however, progress has been made in understanding analogous entropy cones for special subsets of quantum states, in particular for holographic states [1–3], stabilizer states [4], and states with entropies describable by hypergraph models, which define the hypergraph entropy cone [5, 6]. Since holographic entropies are completely captured by graph models [1], it is clear that the holographic entropy cone is contained in the hypergraph cone, a containment which is strict beginning at 3 parties [5]. In this note, we will focus on the nontrivial relationship between the hypergraph and stabilizer entropy cones. In [5], it was shown that these entropy cones coincide for up to and including 4 parties. Though the hypergraph entropy cone was originally studied as a generalization of the holographic one, this observed equivalence with the stabilizer entropy cone naturally led to the conjecture that this relation continues to hold at any party number. Shortly after, [6] showed that the hypergraph entropy cone is contained in the stabilizer entropy cone using

random stabilizer tensor network techniques, thereby proving one direction of the conjectured equivalence. In this work, we will show that the reverse direction of the conjecture is false; namely, that there exists stabilizer states whose entropies do not admit a hypergraph model representation. Given the result of [6] and the fact that some stabilizer entropies cannot be realized by hypergraph models, one is able to conclude that the hypergraph entropy cone in fact lies strictly inside the stabilizer one for 5 and more parties.

The structure of this note is as follows. In Section 2, we review relevant aspects of what is known about entanglement entropies of stabilizer states, with a focus on facts relating stabilizer state entropies to those of the graph states of [7]. In Section 3, we review the results of [5, 6], presenting hypergraph models of entanglement and the associated method for proving entropy inequalities. In addition, we state and prove several results regarding hypergraph minimum cuts of minimal vertex size, which generalize the notion of entanglement wedges in holography. This allows us to prove a hypergraph version of entanglement wedge nesting and to better characterize the behavior of hypergraph entropies. In Section 4, we construct hypergraph representations of the entropies of all graph states for up to six qubits save one. To declare the entropies of the offending graph state non-realizable, we prove novel hypergraph entropy inequalities that exclude it, thus demonstrating strict containment of the hypergraph entropy cone inside the stabilizer one in combination with [6]. Additionally, we use hypergraph-realizable entropies to tighten the known gap between the stabilizer and quantum linear rank cones. We end in Section 5 with some concluding remarks and comments on potential future directions.

## 2   Stabilizer States

Stabilizer states are an important class of quantum states that make frequent appearances across the quantum information literature [8]. Such states derive their name from their efficient description: instead of specifying a state vector, a stabilizer state can instead be specified as the unique simultaneous eigenvector with eigenvalue $+1$ of a given list of operators built out of tensor products of Pauli matrices. Equivalently, they can be described as the set of all states that can be generated by the action of Hadamard, CNOT and phase gates on the all-zeroes state $|0\rangle^{\otimes n}$ for any $n \in \mathbb{Z}_+$.

As a result of such a description, the entanglement structure of stabilizer states is much better understood than that of generic quantum states. Consider the von Neumann entropy,

$$S_I = - \operatorname{Tr} \rho_I \log \rho_I, \tag{2.1}$$

where $\rho_I$ is the reduced density matrix obtained from the $n$-party state $\rho_{1,\dots,n}$ by tracing over all parties $i \notin I \subseteq [n] \equiv \{1, \dots, n\}$. While the entanglement of generic 4-party states $\rho_{ABCD}$ is poorly understood for an arbitrary quantum state, the entropy cone for 4-party stabilizer states has been completely characterized [4]. In particular, it is bounded by the Ingleton inequality, which is not obeyed by general quantum states and reads [4, 9]

$$I(A:B|C) + I(A:B|D) + I(C:D) - I(A:B) \geq 0, \tag{2.2}$$

where $I(A : B) = S_A + S_B - S_{AB}$ denotes the mutual information between $A$ and $B$, and $I(A : B|C) = S_{AC} + S_{BC} - S_{ABC} - S_C$ is the conditional mutual information between $A$ and $B$ conditioned on $C$.

A useful way of constructing qubit stabilizer states is via graphs, which give rise to the notion of *graph states* as described in [7]. Importantly, these graph states are distinct from the graph and hypergraph models of entanglement that we discuss in the other parts of this note. Given a graph $G = (V, E)$ with vertex set $V$ and edge set $E \subset V \times V$, the corresponding graph state is prepared as follows: at every vertex $x \in V$, place a qubit initialized to $|+\rangle = \frac{1}{\sqrt{2}}(|0\rangle + |1\rangle)$. For every edge in $(x, y) \in E$, apply a controlled-$Z$ gate $U_{x,y}$ between the two qubits associated to the two vertices in the edge. The resulting graph state $|G\rangle$ takes the following form:

$$|G\rangle = \prod_{(x,y) \in E} U_{x,y} |+\rangle^{\otimes V}. \tag{2.3}$$

Note that $|+\rangle$ states are symmetric with respect to the target and control of controlled-$Z$ gates, and such gates applied to different qubits commute, implying that the order of application is immaterial.

From the definition, we see that every graph state is a stabilizer state, as controlled-$Z$ gates are equivalent to a combination of CNOT and Hadamard gates. However, the surprisingly useful aspect of these graph states is that they fully characterize the set of stabilizer entropies [7]:

**Theorem 2.1.** *Every stabilizer state is equivalent to a graph state, up to local unitary gates.*

More explicitly, this means that given a stabilizer state $|\psi\rangle$ on $n$ qubits, there is a set of single-qubit unitaries $\{U_1, \ldots, U_n\}$ such that

$$|G\rangle = U_1 \otimes \ldots \otimes U_n |\psi\rangle, \tag{2.4}$$

where $|G\rangle$ is a graph state constructed as in (2.3). Because the von Neumann entropy is invariant under the action of local unitaries, it follows that the entropies of any stabilizer state can be realized by an entropically equivalent graph state related to it by (2.4).

## 3 Hypergraph Entropies

In this section, we first briefly review the hypergraph models of entanglement that were introduced and first studied in [5]. We also highlight the existence of a "contraction map" method for proving entropy inequalities on hypergraph models, which was described in detail in [5] generalizing tools originally developed in [1]. Readers familiar with this material may skip ahead to Section 3.2, where we discuss aspects of hypergraph minimum cuts and prove some properties they obey which generalize ideas in holography such as entanglement wedge nesting.

### 3.1 Hypergraph Models

A *graph* is defined by a pair $G = (V, E)$, consisting of a vertex set $V$ and an edge set $E \subseteq V \times V$, where each edge is a pair of vertices. In the present context all graphs are taken to be undirected, thus consisting of undirected edges given by unordered vertices. A *hypergraph* is a generalization of a graph in which edges are promoted to hyperedges, which are arbitrary subsets of vertices $e \subseteq V$ of cardinality $|e| \geq 2$. An edge of cardinality $k \geq 2$ vertices is called a *k-edge*, and a hypergraph consisting solely of trivial edges and $k$-edges is called a *k-uniform* (hyper)graph. Denoting the power set of $V$ by $\wp(V)$, one thus now upgrades $E$ to a hyperedge set $E \subseteq \wp(V)$. Hyperedges can be assigned capacities or weights via a map $\omega : E \rightarrow \mathbb{R}_+$ to define a weighted hypergraph in much the same way as it is done for standard graphs .We define the *rank* of a hypergraph as the largest cardinality of hyperedges of non-zero weight in the hypergraph. A hypergraph of rank $k$ will be called a *k-(hyper)graph*.

A *hypergraph model* for $n$ parties is simply a weighted hypergraph with the following additional structure: a special subset of vertices $\partial V \subseteq V$, referred to as *external*, are assigned to quantum parties via a *coloring*, i.e. a surjective map $\partial V \rightarrow [n+1]$ where each $i \in [n]$ corresponds to one party and the $(n+1)^{\text{th}}$ color accounts for the purifier. The remaining vertices $V \setminus \partial V$ are generally called *internal* and are responsible for the rich entanglement structure that hypergraph models are able to encode. In particular, the entropy of a subsystem $I \subseteq [n]$ in a hypergraph is defined as the total weight of the minimum cut, or *min-cut*, that separates the external vertices colored by $I$ from the others in $\partial V$.

Hypergraph models encode the entropic structure of a certain class of quantum states, but otherwise make no assumption about the possible underlying properties of those states. Because of the graphical and combinatorial nature of hypergraph min-cuts, hypergraph entropies turn out to obey certain constraints. These take the form of inequalities which are linear and homogeneous in the entropies, and can be proven using a method based on finding a contraction map. Our later results will use inequalities proved with this technique, which we briefly review. For more details, we refer the interested reader to [5].

A general linear entropy inequality may be written as follows:

$$\sum_{l=1}^{L} \alpha_l S(I_l) \geq \sum_{r=1}^{R} \beta_r S(J_r), \tag{3.1}$$

where $S(I_l)$ is the von Neumann entropy of the $I_l$ party and enters the inequality with coefficient $\alpha_l$, and similarly on the RHS for $S(J_r)$ and $\beta_r$. Note that the LHS has $L$ terms and the RHS has $R$ terms. We now introduce a generalization of the weighted Hamming distance between two bitstrings: a weighted indicator function that quantifies the difference between multiple bitstrings, $x^1, \ldots, x^k$:

$$i_\alpha^k(x^1, \ldots, x^k) = \sum_l \alpha_l i^k(x_l^1, \ldots, x_l^k) \tag{3.2}$$

where $x_j^i$ is the $j$th bit of the $i$th bit string, and the $i^k(x_l^1, \ldots, x_l^k)$ on the RHS is 0 if all the bits $x_l^1, \ldots, x_l^k$ are identical and 1 otherwise. In other words, the RHS is a weighted sum over indicator functions $i^k : \{0,1\}^k \to \{0,1\}$.

We can now state the key results that make up the *proof by contraction* method for hypergraphs – see [5] for more details on these results and their proofs.

**Theorem 3.1.** *Let $f : \{0,1\}^L \to \{0,1\}^R$ be an $i_\alpha^k$-$i_\beta^k$ contraction:*

$$i_\alpha^k(x^1, \ldots, x^k) \geq i_\beta^k(f(x^1), \ldots, f(x^k)) \qquad \forall x^1, \ldots, x^k \in \{0,1\}^L. \tag{3.3}$$

*If $f\left(x^{(i)}\right) = y^{(i)}$ for all $i \in [n+1]$, then (3.1) is a valid inequality on $k$-uniform graphs.*

If such a map $f$ exists for a given inequality and a certain class of graphs, we say that the inequality *contracts* on those graphs. Although this theorem only applies to $k$-uniform graphs, it turns out to be sufficient to prove validity of an inequality for all hypergraphs with hyperedges of cardinality $k$ or less:

**Corollary 3.1.1.** *If (3.1) contracts on $k$-uniform graphs, then it contracts on $k$-graphs.*

It also turns out to be possible to upper-bound the rank of hypergraphs on which one needs to prove validity of an inequality in order to be guaranteed its validity on hypergraphs with hyperedges of arbitrary cardinality:

**Proposition 3.1.1.** *If (3.1) contracts on $R$-graphs, then it is a valid inequality on all hypergraphs of finite rank.*

These results provide a recipe for proving linear entropy inequalities on hypergraphs by contraction. In particular, one attempts to explicitly construct a map $f$ satisfying the above contraction conditions on hypergraph models of rank $R$. This involves the computational challenge of finding a set of $2^L$ bitstrings of length $R$ for the images of $f$ compatible with all inequalities specified by Theorem 3.1 Our main result of strict containment comes from such a provable inequality that is violated for a particular stabilizer state.

## 3.2 Properties of Hypergraph Min-Cuts

Ultimately, our proof of strict containment of the hypergraph entropy cone within the stabilizer entropy cone will rely on proving additional hypergraph entropy inequalities that are not obeyed by specific stabilizer states. In parallel, we also attempted to explicitly prove the impossibility of reproducing a given entropy vector via a consistent hypergraph. While this method was ultimately insufficient, we provide here some results from the pursuit of this direction that may find interpretation in the context of generalized versions of holography. Since graphs are a special case of hypergraphs, these will also apply as statements in the holographic limit. The statements we prove can be understood as straightforward extensions of min-cut properties in standard graphs, cf. [10, 11].

It is convenient at this point to introduce some notation and definitions which will be consistently used throughout this section. Recall that a cut is a subset of vertices $W \subseteq V$ which defines a bipartition of the vertex set into two disjoint subsets by $V = W \cup W^{\complement}$. Associated to it, one defines a hyperedge cut $C(W) \subseteq E$ consisting of the subset of hyperedges containing at least one vertex in both $W$ and $W^{\complement}$. The total weight of all hyperedges in $C(W)$ is then given by the cut function $c(W) \equiv \sum_{e \in C(W)} \omega(e)$. Denote now general subsets of external vertices by $X, Y \subseteq \partial V$. One then defines a min-cut $W_X$ for $X$ as a cut with $W_X \cap \partial V = X$ of minimal total weight. In other words, $W_X$ is a non-necessarily-unique set of vertices containing precisely the external ones in $X \subseteq \partial V$ which minimizes the value of the cut function. For $X$ and $Y$ min-cuts we will use the notation $x \equiv W_X$ and $y \equiv W_Y$ below.

**Theorem 3.2.** *If $X \cap Y = \emptyset$, then there exist min-cuts $x, y$ such that $x \cap y = \emptyset$.*[1]

*Proof.* Suppose $X \cap Y = \emptyset$ and let $\delta \equiv x \cap y$. Because $X$ and $Y$ are disjoint, $x \setminus \delta$ is also a cut for $X$ and $c(x) \leq c(x \setminus \delta)$ by the min-cut assumption. The analogous cuts for $Y$ are just $y$ and $y \setminus \delta$. Consider now just the set $x \cup y$ and notice how all previous cuts partition it. In particular, one easily sees that $y \setminus \delta$ partitions $x \cup y$ in precisely the same way as $x$ and, similarly, $y$ partitions $x \cup y$ in the same way as $x \setminus \delta$. Therefore, if $c(x) \leq c(x \setminus \delta)$, then $c(y) \geq c(y \setminus \delta)$. However, because $y$ is a min-cut, we also have $c(y) \leq c(y \setminus \delta)$, implying that $c(y) = c(y \setminus \delta)$. We can then choose $y \setminus \delta$ to be the min-cut for $Y$, which manifestly has empty intersection with $x$. □

This form of disjoint complementarity of the min-cuts of disjoint subsets of external vertices immediately leads to the following result on nesting of hypergraph min-cuts:

**Theorem 3.3.** *If $Y \subset X$, then there exist min-cuts $x, y$ such that $y \subset x$.*

*Proof.* If $Y \subset X$, then $Z \equiv X^{\complement}$ satisfies $Z \cap Y = \emptyset$. Applying Theorem 3.2, there exists min-cuts $y, z$ respectively for $Y, Z$ such that $y \cap z = \emptyset$. Now note that if $z$ is a min-cut for $Z$, then so is $z^{\complement}$ for $Z^{\complement} = X$.[2] Equivalently, $x \equiv z^{\complement}$ defines a min-cut for $X$, and since $y \cap x^{\complement} = \emptyset$, one has that $y \subset x$, thereby proving the desired result. □

In other words, if the subsets of external vertices are nested, then so are their min-cuts, provided they are unique. If they are not unique, then one is always guaranteed that there exist "minimal" min-cuts which are nested. This is precisely the statement of entanglement wedge nesting in the holographic context, now generalized to hypergraphs beyond holography.

When attempting to construct a hypergraph that reproduces a particular realizable entropy vector, there is in general a large degree of freedom in how one arranges the internal

---

[1]We note that an analogous result for standard 2-graphs was succinctly proven in [11], Lemma 9, by applying submodularity and symmetry of the graph cut function. Because the hypergraph cut function is also submodular and symmetric, their proof also applies in this case. The proof presented here is essentially an explicit and more intuitive version of theirs.

[2]Here the complement of cuts is taken with respect to the full vertex set, i.e. $z^{\complement} \equiv V \setminus z$, whereas the complement of external vertices is taken only with respect to boundary vertices, i.e. $Z^{\complement} = \partial V \setminus Z$.

vertices. Trying to fix this arbitrariness without loss of generality allows one to constrain the structure of hypergraphs one needs to consider. Put differently, one can always dispense with unnecessary structures, particularly when the entropy vector obeys certain properties. The following simple result shows that no additional internal vertices are needed to account for the min-cut of $X \cup Y$ when its constituents $X$ and $Y$ share no mutual information:

**Theorem 3.4.** If $I_2(X : Y) = 0$, then $xy = x \cup y$ is a min-cut for $X \cup Y$.

*Proof.* In a general hypergraph, $x \cup y$ is clearly a cut for $X \cup Y$ which satisfies $c(x \cup y) \leq c(x) + c(y)$. However the assumption that $I_2(X : Y) = 0$ means that the min-cut $xy$ of $X \cup Y$ obeys $c(xy) = c(x) + c(y)$. It must thus be the case that $c(x \cup y)$ attains its maximum value $c(xy)$, which itself is the min-cut value. It follows that $x \cup y$ is a min-cut for $X \cup Y$. □

Note that the above result is easily understood in terms of the equivalent stabilizer tensor network that realizes the given entropy vector. It is merely the statement that when the mutual information between two subsystems $X$ and $Y$ vanishes, the joint state is a product state, with no correlations between the subsystems.

## 4  Entropy Cones

In this section, we share the main technical result regarding the containment relation between the hypergraph and stabilizer entropy cones. In particular, we show that the former is strictly contained inside the latter by proving entropy inequalities that separate the two cones appropriately. Before discussing this finding, we briefly review the formalism surrounding entropy cones and summarize the state of known results in Table 1.

### 4.1  Preliminaries

Given a quantum state $\rho_{1,\dots,n}$ on $n$ parties, the entanglement entropy of a nonempty subsystem $I \subseteq [n]$ is given by (2.1). With $n$ parties, and excluding the empty set, there are $D \equiv 2^n - 1$ possible such subsystems. The *entropy vector* of the state is given by an ordered collection of these $D$ entropies, and we refer to the positive octant $\mathbb{R}_+^D$ to which they belong as *entropy space*. The collection of all entropy vectors of a given subset of quantum states defines what is called an *entropy cone*. When this cone is polyhedral, it can be fully specified in two equivalent ways: in terms of its outermost entropy vectors, called *extreme rays*, or in terms of the tightest inequalities that define its supporting hyperplanes, known as *facets*.

The following two universal quantum inequalities,

$$S_A + S_B \geq S_{AB}, \tag{4.1}$$

$$S_{AB} + S_{BC} \geq S_{ABC} + S_B, \tag{4.2}$$

respectively known as *subadditivity* (SA) and *strong subadditivity* (SSA), completely characterize the quantum entropy cone for up to $n = 3$. That is, a vector in $\mathbb{R}_+^7$ satisfies the above inequalities if and only if there exists a tripartite state $\rho_{ABC}$ such that its subsystem

entropies match the given vector. While the entropy cone for 4 parties and higher is unknown for generic quantum states, complete descriptions have been obtained for holographic, stabilizer and hypergraph entropies at this party number (see Table 1 for a summary of results).

The holographic entropy cone was shown to be equivalent to the graph entropy cone for any number of parties [1], and is known completely for up to $n = 5$ [2, 3]. Due to the holographic inequality of monogamy of mutual information (MMI), this cone is already strictly contained inside the stabilizer entropy cone for $n = 3$. The stabilizer entropy cone is known for up to $n = 4$ [4, 9] and, up to this number of parties, it is completely characterized by SA, SSA, and the Ingleton inequality (2.2). This cone is equivalent to the 4-party quantum linear rank (QLR) cone [5], defined by combining all linear rank inequalities that relate the ranks of subspaces of vector spaces [12], but including weak monotonicity instead of classical monotonicity.[3] Furthermore, it was shown in [5] that the entropy cone of 4-party stabilizer states is also precisely equivalent to the entropy cone of 4-party hypergraphs. However, it remained as an open question whether the equivalence between the stabilizer, QLR, and hypergraph cones continued to hold for higher party numbers. As will be shown in Section 4.3, the answer to this question is negative: there exists a class of stabilizer entropy vectors that cannot be realized by hypergraphs. Combined with [6], which shows that hypergraph entropy vectors are stabilizer-realizable for all party numbers, this allows us to conclude that the hypergraph entropy cone is strictly contained inside the stabilizer entropy cone starting precisely at $n = 5$ parties.

## 4.2 Hypergraph, Stabilizer and Linear Rank Cones

It is challenging to study the relation between the hypergraph and stabilizer cones for $n \geq 5$ because neither of them is known completely. Nevertheless, substantial progress can be made in the analysis of these objects by exploring the intimately related structure of cones bounded by linear rank inequalities [12, 13].

In what follows, it will be useful to consider not only the QLR cone introduced above, but also its cousin, the classical linear rank (CLR) cone, which includes classical monotonicity as an additional facet (thus making weak monotonicity a redundant inequality). The reason for this is that, while an extreme ray description of the QLR cone could not be obtained due to computational limitations, [5] managed to show that all extreme rays of the CLR cone are realizable by hypergraphs. Note that even though the facet description of the CLR and QLR cones differ only in that the latter does not include classical monotonicity as a facet, many of the extreme rays of the CLR cone cease to be extreme in the larger QLR cone. Indeed, one finds that out of the 162 orbits of extreme rays of the former, only 40 remain extreme in the latter. Nevertheless, the hypergraph-realizable extreme rays of the CLR cone suffice to show that all 31 QLR inequalities for $n = 5$ are facets of the hypergraph cone. By the result of [6], this implies that all QLR inequalities are facets of the stabilizer cone as well. To our

---

[3]The Ingleton inequality is the only new such inequality for $n = 4$ apart from SA and SSA. See [5] for more details on the QLR cone and [13] for explicit linear rank inequalities for up to $n = 5$.

| Entropy Cone | $n = 2$ | $n = 3$ | $n = 4$ | $n = 5$ | $n = 6$ |
|---|---|---|---|---|---|
| Quantum | SA, SSA | – | ? | ? | ? |
| Stabilizer | SA, SSA | – | Ingleton | QLR+? | ? |
| Hypergraph | SA, SSA | – | Ingleton | QLR+(4.4)+? | ? |
| Holographic | SA, SSA | MMI | – | 5 in [1–3] | [14] |

**Table 1**: Characterization in terms of inequalities of the entropy cones discussed here. Cones are ordered by inclusion from the largest to the smallest. In the columns, we list the inequalities that show up at increasing $n$, and use "–" to indicate no new inequalities for a given party number. Since every cone in each row is a subset of the previous one, all inequalities are inherited (possibly redundantly) for a given column. Cones with a currently incomplete description above some $n$ include question marks. All listed inequalities are facets, except for (4.4), which separates the stabilizer and hypergraph cones, but is not expected to be a facet of the latter because it is not balanced.

knowledge, this is the first time that the QLR inequalities have been checked to be facets for the stabilizer entropy cone[4] for all $n \leq 5$, thus extending the boundedness result of [4].

Not only have hypergraphs allowed us to prove tightness of the QLR inequalities that bound the stabilizer entropy cone, but they also let us generate some results about the reach of stabilizer extreme rays. More precisely, proving whether the QLR inequalities completely define the stabilizer cone would require an explicit realization of all of its extreme rays via stabilizer states. Although such an endeavour is beyond the scope of this note, we do highlight here that the realizability of all CLR extreme rays via hypergraphs already provides a nontrivial inner bound to the stabilizer cone [6]. In fact, this inner bound reaches the QLR facets at the 40 orbits of extreme rays that the CLR and QLR share. Remarkably, one can also verify that all of the extreme ray orbits of the holographic entropy cone for 5 parties except for the 18[th] as listed in Table 3 of [2] constitute extreme rays of the QLR cone, and thus of the hypergraph and stabilizer ones as well. Of these, all but one are distinct from the 40 that come from the CLR cone, thus revealing 17 additional extreme rays of the QLR cone. Furthermore, from the 19 graph state entropy vectors in Table 2, one finds that only two give new extreme ray orbits of the QLR cone, corresponding to vectors 11 and 15. However, as will be shown below, the latter turns out to not be realizable by hypergraphs. This reveals one key result of this note: we have been able to enumerate 59 extreme rays of the QLR cone which are also extreme rays of the stabilizer cone, and pick out a unique vector 15 in Table 2 as the only extreme ray which is not hypergraph-realizable. While the other 58 are of course extreme rays of the hypergraph cone, this particular one happens to lie strictly outside of it and will provide us with the result that hypergraph entropy vectors are strictly contained inside the stabilizer cone.

---

[4]We thank an anonymous referee for improving our discussion on facets vs tight inequalities.

### 4.3 Strict Containment

The coincidence of the hypergraph entropy cone with the stabilizer one for all $n \leq 4$ that was found in [5] was unexpected. Together with some additional suggestive evidence for $n = 5$, this motivated a conjectured equivalence between the two cones at higher party numbers. Soon thereafter, it was shown in [6] that indeed one of the directions of this conjecture is correct: all hypergraph entropies are stabilizer-realizable, thus proving containment of the hypergraph cone in the stabilizer one. Here we show that, in contrast, the converse does not hold for any $n \geq 5$. In other words, this cone containment is strict, in the sense that there exists stabilizer entropy vectors which cannot be realized by hypergraphs starting at $n = 5$.

This result effectively follows by counterexample; we identify a specific entropy vector (namely, vector 15 in Table 2) which belongs to the stabilizer cone but lies strictly outside of the hypergraph cone. Such a finding is particularly nontrivial because, as previously stressed, neither the stabilizer nor the hypergraph cones are actually known for any $n \geq 5$. Here we guide our search for candidate hypergraph inequalities by focusing on the graph states described previously. In particular, for mixed states on 5 parties, we consider the 6-qubit graph states in Figure 4 of [7], which are themselves all stabilizer states. The entropy vectors of these graph states are listed in order in Table 2. An exhaustive search for hypergraph instantiations of these entropy vectors for up to 5 internal vertices turns out to yield solutions for all but graph 15, as shown in Figure 1. This is suggestive that its entropies are not realizable by hypergraphs, but of course does not provide a definite answer by itself. To make this conclusive, we proceed by searching for an inequality that rules it out and can be proven to be true for all hypergraphs via a contraction map.

Reproducing from Table 2, the entropy vector of graph state 15 reads

$$\vec{S}(G_{15}) = (1\,1\,1\,1\,1\,;2\,2\,2\,2\,2\,1\,2\,2\,1\,2\,;2\,2\,2\,2\,2\,2\,2\,2\,2\,2\,;2\,2\,2\,2\,1\,;1)\,. \tag{4.3}$$

Hence we search for an entropy inequality of the form $\vec{Q} \cdot \vec{S} \geq 0$ that is valid on any hypergraph entropy vector $\vec{S} \in \mathbb{R}_+^D$ but violated by (4.3), i.e. such that $\vec{Q} \cdot \vec{S}(G_{15}) < 0$. Geometrically, the vector $\vec{Q} \in \mathbb{R}^D$ which defines such an inequality corresponds to the vector normal to the hyperplane $\vec{Q} \cdot \vec{S} = 0$ in entropy space. As such, what we would like to obtain is a hyperplane that excises the vector $\vec{S}(G_{15})$ away from the half-space $\vec{Q} \cdot \vec{S} \geq 0$ in which the hypergraph entropy cone is contained. Although the latter is not known completely, a substantial partial description of it can be attained by collecting the results in [2, 5] and Figure 1. In particular, these show that, for 5 parties, all extreme rays of the holographic entropy cone [2], all extreme rays of the CLR cone [5], and all entropy vectors in Table 2 corresponding to 6-qubit graph states [7], with the only exception of 15, are hypergraph-realizable. Therefore, a necessary condition for a vector $\vec{Q}$ to define the desired hyperplane is that it obeys $\vec{Q} \cdot \vec{S} \geq 0$ for all of these vectors $\vec{S}$, known to have hypergraph realizations. By simultaneously demanding that $\vec{Q} \cdot \vec{S}(G_{15}) < 0$, one can search for $\vec{Q}$ solutions and hope that at least one be a valid inequality on hypergraphs and provable by the method of contraction described in [5].

Precisely as desired, it was possible to obtain some candidate $\vec{Q}$ solutions which could then be conclusively proven to define valid inequalities on arbitrary hypergraphs. Among

1. $(1\,1\,0\,0\,0;0\,1\,1\,1\,1\,1\,1\,0\,0\,0;0\,0\,0\,1\,1\,1\,1\,1\,1\,0;0\,0\,0\,1\,1;0)$
2. $(1\,1\,1\,0\,0;1\,1\,1\,1\,1\,1\,1\,1\,1\,0;0\,1\,1\,1\,1\,1\,1\,1\,1\,1;0\,0\,1\,1\,1;0)$
3. $(1\,1\,1\,1\,0;1\,1\,1\,1\,1\,1\,1\,1\,1\,1;1\,1\,1\,1\,1\,1\,1\,1\,1\,1;0\,1\,1\,1\,1;0)$
4. $(1\,1\,1\,1\,0;1\,2\,2\,1\,2\,2\,1\,1\,1\,1;1\,1\,1\,1\,2\,2\,1\,2\,2\,1;0\,1\,1\,1\,1;0)$
5. $(1\,1\,1\,1\,1;1\,1\,1\,1\,1\,1\,1\,1\,1\,1;1\,1\,1\,1\,1\,1\,1\,1\,1\,1;1\,1\,1\,1\,1;0)$
6. $(1\,1\,1\,1\,1;1\,2\,2\,1\,2\,2\,1\,1\,2\,2;2\,2\,1\,1\,2\,2\,1\,2\,2\,1;1\,1\,1\,1\,1;0)$
7. $(1\,1\,1\,1\,1;1\,2\,2\,2\,2\,2\,2\,2\,2\,1;1\,2\,2\,2\,2\,2\,2\,2\,2\,1;1\,1\,1\,1\,1;0)$
8. $(1\,1\,1\,1\,1;2\,2\,2\,2\,2\,2\,2\,2\,2\,2;2\,2\,2\,2\,2\,2\,2\,2\,2\,2;1\,1\,1\,1\,1;0)$
9. $(1\,1\,1\,1\,1;1\,1\,1\,1\,1\,1\,1\,1\,1\,1;1\,1\,1\,1\,1\,1\,1\,1\,1\,1;1\,1\,1\,1\,1;1)$
10. $(1\,1\,1\,1\,1;1\,1\,2\,2\,1\,2\,2\,2\,2\,1;1\,2\,2\,2\,2\,1\,2\,2\,1\,1;2\,2\,1\,1\,1;1)$
11. $(1\,1\,1\,1\,1;1\,2\,2\,2\,2\,2\,2\,1\,1\,1;2\,2\,2\,2\,2\,2\,2\,2\,2\,1;2\,2\,2\,1\,1;1)$
12. $(1\,1\,1\,1\,1;1\,2\,2\,2\,2\,2\,2\,2\,2\,1;2\,2\,2\,2\,2\,2\,2\,2\,2\,1;2\,2\,2\,1\,1;1)$
13. $(1\,1\,1\,1\,1;1\,2\,2\,2\,2\,2\,2\,2\,2\,1;1\,2\,2\,3\,3\,2\,3\,3\,2\,1;2\,2\,1\,2\,2;1)$
14. $(1\,1\,1\,1\,1;1\,2\,2\,2\,2\,2\,2\,2\,2\,2;1\,2\,2\,2\,3\,3\,2\,3\,3\,2;1\,2\,2\,2\,2;1)$
15. $(1\,1\,1\,1\,1;2\,2\,2\,2\,2\,1\,2\,2\,1\,2;2\,2\,2\,2\,2\,2\,2\,2\,2\,2;2\,2\,2\,2\,1;1)$
16. $(1\,1\,1\,1\,1;1\,2\,2\,2\,2\,2\,2\,2\,2\,1;2\,2\,2\,3\,3\,2\,3\,3\,2\,2;2\,2\,1\,2\,2;1)$
17. $(1\,1\,1\,1\,1;2\,2\,2\,2\,2\,2\,2\,2\,2\,2;2\,2\,2\,2\,3\,3\,2\,3\,3\,2;1\,2\,2\,2\,2;1)$
18. $(1\,1\,1\,1\,1;2\,2\,2\,2\,2\,2\,2\,2\,2\,2;2\,3\,3\,3\,2\,3\,2\,3\,3\,2;2\,2\,2\,2\,2;1)$
19. $(1\,1\,1\,1\,1;2\,2\,2\,2\,2\,2\,2\,2\,2\,2;3\,3\,3\,3\,3\,3\,3\,3\,3\,3;2\,2\,2\,2\,2;1)$

**Table 2**: Entropy vectors for each of the 19 graph states on 6 qubits in Figure 4 of [7]. Entropies are listed in lexicographical order, with subsystems of increasing number of parties separated by a semicolon.

.

various such inequalities which we managed to prove by contraction, we found

$$S_{BD}+S_{CE}+S_{ABC}+S_{ABE}+2S_{ACD}+S_{BCDE}-2S_{AB}-S_{AC}-S_{BE}-S_{CD}-S_{ABCD} \geq 0, \ (4.4)$$

which constitutes a valid inequality on hypergraphs. A contraction map which proves (4.4) can be found in Appendix A. The LHS of the inequality evaluates to $-1$ on (4.3), thereby ruling out $\vec{S}(G_{15})$ as possibly being realizable by hypergraphs. We obtained 10 other symmetry inequivalent inequalities which are altogether non-redundant and similarly exclude $\vec{S}(G_{15})$ from the hypergraph cone. Since their structure does not seem to be particularly illuminating, we have decided not to include them in this note, but they are available to the interested reader upon request. The existence of (4.4) and other inequalities bounding the hypergraph cone implies that some subspace of vectors of the stabilizer cone are inaccessible to hypergraphs. Using the containment result of [6], the validity of (4.4) thus proves strict containment of the hypergraph cone inside the stabilizer cone for all $n \geq 5$.

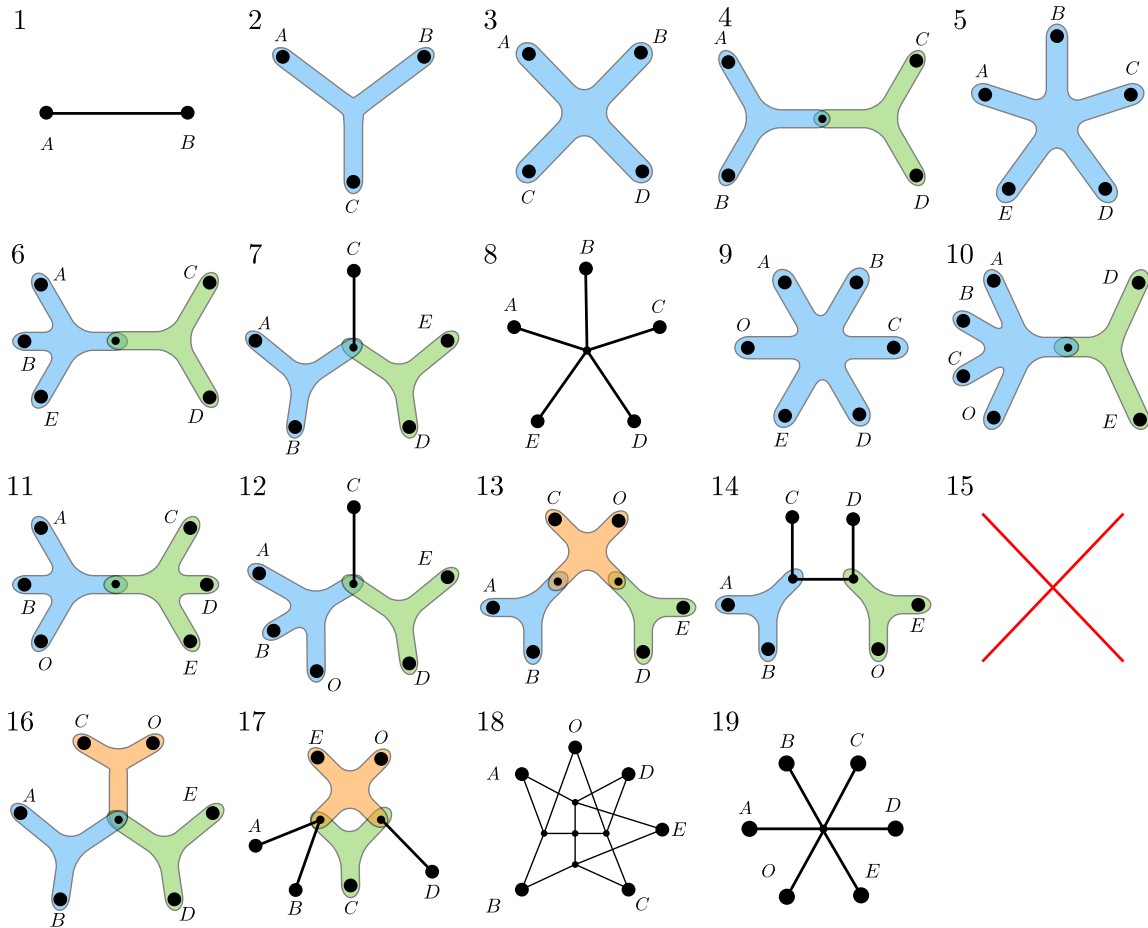

**Figure 1**: Hypergraph realizations of the entropy vectors listed in Table 2. All edges and hyperedges have unit weight. External vertices are labelled by letters in alphabetic order, with $O$ reserved for the purifier. Internal vertices are represented as unlabelled dots. Graph 15 does not have a hypergraph realization.

It is worth noting that, while valid, it is unlikely that the inequality in (4.4) defines a facet of the hypergraph cone. Put differently, whatever the facets of the latter are (which include all 5-party QLR inequalities as explained in Section 4.2), we expect them to yield some inequalities stronger than (4.4). The reason for this is that (4.4) is not symmetry-related to any balanced inequality,[5] and general symmetry arguments suggest that all hypergraph facets should have balanced representatives in their symmetry orbits.[6] From a more physical standpoint, and assuming that hypergraphs represent an interesting class of quantum entropy

---

[5]An entropy inequality is balanced in some party $X$ if the sum of the coefficients carried by entropy terms involving $X$ is zero. One easily checks that (4.4) fails to be balanced because of $E$.

[6]This feature of not having balanced representatives also extends to all other inequalities found thus far valid for the hypergraph entropy cone that exclude $\vec{S}(G_{15})$.

rays, we would expect these inequalities to also define meaningful boundaries in entropy space. In the context of QFT, balance is a natural requisite for a quantity to be finite and scheme-independent upon regularization of von Neumann entropies [15]. Hence failure to obey balance suggests that the saturation of the inequality by entropy rays is not scheme-independent, which makes it less physically meaningful.

## 5    Discussion

Given the inequivalence of the hypergraph entropy cone to both the quantum and stabilizer entropy cones, it is tantalizing to consider whether further generalizations of it could coincide with either of these cones at some, or perhaps all higher party numbers. One potential avenue of exploration would be to topologically link or braid otherwise disjoint graphs or hypergraphs, and to generalize the notion of the cut entropy of a given external vertex set to the min-cut that removes not only all hyperedge paths between the chosen external vertices and the others, but also all links that connect them. It is clear that all hypergraph cut entropies are a special case of this generalization, corresponding to the case in which one has a single link. Including such topological structure could potentially endow the cut problem with additional flexibility to realize entropy vectors that a single hypergraph could not. Interestingly, the addition of such topological structure to hypergraph models of entanglement could tie back to the topological considerations of [16–18] in relating link invariants to entanglement entropy.

Although we have shown strict containment of the hypergraph entropy cone within the stabilizer one starting at $n = 5$, this note still leaves the full description of the hypergraph cone for 5 and higher party numbers as an open problem. More generally, given this strict containment relation, it would be interesting to classify the subset of stabilizer states that are hypergraph-realizable, and clarify the properties that separate them from states that are not. In other words, we would like to better understand if there is a fundamental connection between hypergraphs and some class of stabilizer states. Since stabilizer entropies can be understood in terms of algebraic [4, 19] and phase space formalisms [9], it would be interesting to investigate whether there is a precise relation between entropies in these contexts and in our hypergraph models of entanglement for states with hypergraph-realizable entropies. It is further possible that, for such states, the hypergraph construction can give an efficient algorithm for the computation of entanglement entropies of the states in question, something which is useful for efficient resource estimation.

Finally, it is worth noting that currently there is significant evidence for and none against the coincidence of the stabilizer and QLR cones. This is also an interesting possibility worthy of further exploration, possibly with the tools developed for hypergraph entropies or generalizations thereof.

## Acknowledgements

We thank Nils Baas and Michael Walter for useful discussions and comments. N.B. is supported by the National Science Foundation under grant number 82248-13067-44-PHPXH, by the Department of Energy under grant number DE-SC0019380, and by the Computational Science Initiative at Brookhaven National Laboratory. S.H.C. is supported by "la Caixa" Foundation (ID 100010434) under fellowship LCF/BQ/AA17/11610002 and by the National Science Foundation under grant number PHY-1801805. V.P.S. gratefully acknowledges support by the NSF GRFP under grant number DGE-1752814. We would also like to extend a special thank you to essential workers and their families during this time of crisis.

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

# A Contraction Map Proof

In Section 4.3, we claimed inequality (4.4) to be valid for hypergraphs, thereby ruling out the stabilizer ray 15 in (4.3) from the hypergraph entropy cone. Here, we provide a tabular representation of a contraction map which proves validity of inequality (4.4) on hypergraphs according to the results at the end of Section 3.1. See [5] for more details.

| | BD | CE | ABC | ABE | 2ACD | BCDE | AB | AB | AC | BE | CD | ABCD |
|---|---|---|---|---|---|---|---|---|---|---|---|---|
| O | 0 | 0 | 0 | 0 | 0 | 0 | 0 | 0 | 0 | 0 | 0 | 0 |
| A | 0 | 0 | 1 | 1 | 1 | 0 | 1 | 1 | 1 | 0 | 0 | 1 |
| B | 1 | 0 | 1 | 1 | 0 | 1 | 1 | 1 | 0 | 1 | 0 | 1 |
| C | 0 | 1 | 1 | 0 | 1 | 1 | 0 | 0 | 1 | 0 | 1 | 1 |
| D | 1 | 0 | 0 | 0 | 1 | 1 | 0 | 0 | 0 | 0 | 1 | 1 |
| E | 0 | 1 | 0 | 1 | 0 | 1 | 0 | 0 | 0 | 1 | 0 | 0 |
| | 0 | 0 | 0 | 0 | 0 | 1 | 0 | 0 | 0 | 0 | 0 | 1 |
| | 0 | 0 | 1 | 0 | 0 | 0 | 0 | 0 | 0 | 0 | 0 | 1 |
| | 0 | 0 | 0 | 1 | 0 | 0 | 0 | 0 | 0 | 0 | 0 | 1 |
| | 0 | 0 | 0 | 0 | 1 | 0 | 0 | 0 | 1 | 0 | 0 | 1 |
| | 0 | 0 | 0 | 0 | 1 | 1 | 0 | 0 | 0 | 0 | 0 | 1 |
| | 0 | 0 | 0 | 1 | 0 | 1 | 0 | 0 | 0 | 1 | 0 | 1 |
| | 0 | 0 | 1 | 0 | 1 | 0 | 0 | 1 | 1 | 0 | 0 | 1 |
| | 0 | 0 | 1 | 1 | 0 | 0 | 0 | 1 | 0 | 0 | 0 | 1 |
| | 0 | 0 | 1 | 0 | 0 | 1 | 0 | 0 | 0 | 1 | 0 | 1 |
| | 0 | 0 | 1 | 1 | 0 | 1 | 0 | 1 | 0 | 1 | 0 | 1 |
| | 0 | 0 | 1 | 0 | 1 | 1 | 0 | 0 | 1 | 0 | 0 | 1 |
| | 0 | 0 | 0 | 1 | 1 | 0 | 0 | 1 | 1 | 0 | 0 | 1 |
| | 1 | 0 | 0 | 0 | 0 | 1 | 0 | 0 | 0 | 1 | 0 | 1 |
| | 1 | 0 | 0 | 0 | 0 | 0 | 0 | 0 | 0 | 0 | 0 | 1 |
| | 0 | 1 | 0 | 0 | 0 | 1 | 0 | 0 | 0 | 0 | 0 | 0 |
| | 0 | 1 | 0 | 0 | 0 | 0 | 0 | 0 | 0 | 0 | 0 | 0 |
| | 1 | 0 | 0 | 0 | 1 | 0 | 0 | 0 | 0 | 0 | 0 | 1 |
| | 0 | 1 | 0 | 1 | 0 | 0 | 0 | 0 | 0 | 0 | 0 | 0 |
| | 1 | 0 | 1 | 0 | 0 | 1 | 0 | 1 | 0 | 1 | 0 | 1 |
| | 1 | 0 | 1 | 0 | 0 | 0 | 0 | 0 | 0 | 1 | 0 | 1 |
| | 0 | 1 | 1 | 0 | 0 | 1 | 0 | 0 | 0 | 0 | 0 | 1 |
| | 0 | 1 | 1 | 0 | 0 | 0 | 0 | 0 | 0 | 0 | 0 | 1 |
| | 1 | 0 | 0 | 1 | 0 | 0 | 0 | 0 | 0 | 1 | 0 | 1 |
| | 1 | 0 | 0 | 1 | 0 | 1 | 0 | 1 | 0 | 1 | 0 | 1 |
| | 1 | 0 | 1 | 1 | 0 | 0 | 0 | 1 | 0 | 1 | 0 | 1 |
| | 0 | 1 | 0 | 0 | 1 | 0 | 0 | 0 | 0 | 0 | 0 | 1 |
| | 0 | 1 | 0 | 0 | 1 | 1 | 0 | 0 | 0 | 0 | 1 | 1 |
| | 0 | 1 | 1 | 0 | 1 | 0 | 0 | 0 | 1 | 0 | 0 | 1 |
| | 0 | 0 | 0 | 1 | 1 | 1 | 0 | 0 | 1 | 0 | 0 | 1 |
| | 0 | 0 | 1 | 1 | 1 | 1 | 0 | 1 | 1 | 0 | 0 | 1 |
| | 0 | 1 | 1 | 1 | 0 | 1 | 0 | 0 | 0 | 1 | 0 | 1 |
| | 0 | 1 | 1 | 1 | 0 | 0 | 0 | 0 | 0 | 0 | 0 | 1 |
| | 1 | 0 | 1 | 0 | 1 | 1 | 0 | 0 | 0 | 0 | 0 | 1 |
| | 1 | 0 | 1 | 0 | 1 | 0 | 0 | 0 | 1 | 0 | 0 | 1 |
| | 1 | 0 | 0 | 1 | 1 | 0 | 0 | 0 | 1 | 0 | 0 | 1 |
| | 1 | 0 | 0 | 1 | 1 | 1 | 0 | 0 | 0 | 0 | 0 | 1 |
| | 1 | 0 | 1 | 1 | 1 | 0 | 0 | 1 | 1 | 0 | 0 | 1 |
| | 1 | 0 | 1 | 1 | 1 | 1 | 0 | 1 | 0 | 0 | 0 | 1 |
| | 0 | 1 | 0 | 1 | 1 | 0 | 0 | 0 | 1 | 0 | 0 | 1 |
| | 0 | 1 | 0 | 1 | 1 | 1 | 0 | 0 | 0 | 0 | 0 | 1 |
| | 0 | 1 | 1 | 1 | 1 | 0 | 0 | 1 | 1 | 0 | 0 | 1 |
| | 0 | 1 | 1 | 1 | 1 | 1 | 0 | 0 | 1 | 0 | 0 | 1 |
| | 1 | 1 | 0 | 0 | 0 | 1 | 0 | 0 | 0 | 0 | 0 | 1 |
| | 1 | 1 | 0 | 0 | 0 | 0 | 0 | 0 | 0 | 0 | 0 | 0 |
| | 1 | 1 | 1 | 0 | 0 | 1 | 0 | 0 | 0 | 1 | 0 | 1 |
| | 1 | 1 | 1 | 0 | 0 | 0 | 0 | 0 | 0 | 0 | 0 | 1 |
| | 1 | 1 | 0 | 1 | 0 | 0 | 0 | 0 | 0 | 1 | 0 | 1 |
| | 1 | 1 | 1 | 1 | 0 | 1 | 0 | 1 | 0 | 1 | 0 | 1 |
| | 1 | 1 | 1 | 1 | 0 | 0 | 0 | 0 | 0 | 1 | 0 | 1 |
| | 1 | 1 | 0 | 0 | 1 | 0 | 0 | 0 | 0 | 0 | 0 | 0 |
| | 1 | 1 | 0 | 0 | 1 | 1 | 0 | 0 | 0 | 0 | 1 | 1 |
| | 1 | 1 | 1 | 0 | 1 | 1 | 0 | 0 | 0 | 0 | 0 | 1 |
| | 1 | 1 | 1 | 0 | 1 | 0 | 0 | 0 | 0 | 0 | 0 | 1 |
| | 1 | 1 | 0 | 1 | 1 | 0 | 0 | 0 | 0 | 0 | 0 | 0 |
| | 1 | 1 | 0 | 1 | 1 | 1 | 0 | 0 | 0 | 0 | 0 | 1 |
| | 1 | 1 | 1 | 1 | 1 | 0 | 0 | 0 | 1 | 0 | 0 | 1 |
| | 1 | 1 | 1 | 1 | 1 | 1 | 0 | 0 | 0 | 0 | 0 | 1 |