# Peer review of "A Gap Between the Hypergraph and Stabilizer Entropy Cones"

_SciPost Physics_

## Round 2 · Referee Report · Anonymous (Referee 1) · 2022-2-17

Strengths

1 - The authors provide a specific extremal ray to show that the hypergraph entropy cone is strictly contained in the stabilizer cone for n=5. This ray takes the form of a graph state, which are a well-studied class of quantum states. They identify a hypergraph entropy cone facet which forms an inequality that is violated by that specific graph state. I think this is a particularly nice way to arrive to their result, given that it takes the form of a well-studied class of states and in form of an explicit example.
2 - They show that all linear rank inequalities form also facets of the stabilizer cone.

Weaknesses

1 - The paper frequently has a type of "story-mode" writing which at times is hard to follow. Examples are the second part of Section 4.1, Section 4.2, and first part of Section 4.3. A more factual and less repetitive style would be helpful, and would probably shorten the manuscript to half its lenght. 2 - Many key steps for main results, such as how 40 out 162 CLR rays remain extreme, how the extreme QLR rays were enumerated, and how exactly Q-solutions were found, are not detailed. 3 - It is unclear how Section 3 fits into the manuscript, the results on entanglement wedge nesting appear out of context. Further results that are not directly related to the claim in the title and for the later development in the manuscript could be moved to the appendix, in particular if they are straightforward extensions for the case of standard graphs as the authors write. 4 - Many of the key statements and definitions are only explained in words but not in formulae. For example, considering the various inclusion relations of the different cones, one really expects to see formulae containing \subseteq and \subsetneq for the cases n=4, n=5. Likewise, the entropy in form of a min-cut is not stated a formula. 5 - The manuscript's key results are not stated in theorem environments, and the proofs are not formulated as classical proofs. All appearing theorems and propositions are from other references. 6 - The authors write that "neither the stabilizer nor the hypergraph cones are actually known for any n ≥ 5." However, [5] and [6] give a complete list of graph states up to n=6 and n=12 qubits, from which the entropy cone can be spanned. 7 - The results of the authors lead to a natural question, not addressed in the discussion: What are hypergraph entropies useful for, if they describe neither holographic nor (as the authors prove) stabilizer states? How could they (as the authors write) be useful for efficient calculations of entropies or for resource estimation, if the min-cut definition relies on an optimization itself, and no physical interpretation is known?

Report

The authors disprove a conjecture which they made in one of their recent papers. The result is novel and non-trivial. However I do see how it meets the acceptance criteria (i.e. how it meets one of the 1.-4. Expectations).

Requested changes

1 - The results apply to qubits only, make this precise. 2 - Make the definition of stabilizer states precise: the stabilizer elements need to commute, -I cannot be part of the stabilizer, and in total 2^n stabilizer elements are needed so that the resulting state is pure. 3 - The authors state that the entanglement of 4-party states is poorly understood. In fact quite a lot is known, even if only considering the two classical papers [2,3]. 4 - It is important to highlight that implicitly, the stabilizer cone is already known up to 12 qubits throught the database [4]. 5 - It would be good to highlight that the von Neumann entropy is a bipartite measure of entanglement only, even if here it is used in a multipartite setting. 6 - Graph states are defined in Eq (2.3), but never used. The Theorem 2.1 is rather well-known property and should not take the place of a theorem, given that it is not used prominently later. 7 - Clarify why inequality 3.1 is grouped by L and R? 8 - Section 3: highlight that hypergraph entropies are unrelated to hypergraph states. 9 - Theorem 3.1: what is a i_a^k-i_b^k contraction? what is y^{(i)}? 10 - Proposition 3.1.1: "contracts on R-graphs". Why is capital R is used and not say k? It seems unrelated to the previous use of L and R. 11 - It would be useful to have figures to understand the sets X,Y, dV, V, purifier, entanglement wedge nesting, the sharing of facets of QLR and stabilizer cones. 12 - The definition of min-cut is only in words, state a formula. 13 - The authors define a cut to be a subset of vertices. It is clear that a cut can be associated to one of either subset defined through the cut, but at first reading this is rather confusing. 14 - Theorem 3.2: the symbol for min-cuts changed from capital X,Y to small x,y. 15 - What is the equivalent stabilizer tensor network realizing a given entropy vector? Does there always exist such stabilizer tensor network? 16 - "Entropy space" is defined but not used. 17 - The authors write "Nevertheless, the hypergraph-realizable extreme rays of the CLR cone suffice to show that all 31 QLR inequalities for n=5 are facets of the hypergraph cone". How? And does this mean that the cones coincide, or just that the two cones share some facets? Mention this in the text, and a figure would help. 18 - Furthermore, "By the result of [6], this implies that all QLR inequalities are facets of the stabilizer cone as well." How, and what result of [6] is used? 19 - The proof of one key result, namely that all QLR inequalities also form facets for the stabilizer cone, is hard to follow. Key steps such as how 40 out 162 rays remain extreme are not detailed. 20 - "Remarkably, one can verify that all of the extreme ray orbits of the holographic entropy cone.." By the wording it is not clear that this is what was done by the authors, and how this was done. 21 - Section 4.3, "mixed states on 5 parties, we consider 6-qubit graph states." Should mention reductions of these states. 22 - An exhaustive search for hypergraphs was made, but this does not provide a definite answer according to the authors. Why not, if the search was exhaustive? 23 - "general symmetry arguments suggest" What are these arguments? 24 - How are the mentioned extreme ray orbits of the holographic entropy generated and compared to those of the QLR cone?

[1] Ning Bao, Newton Cheng, Sergio Hernández-Cuenca, Vincent P. Su, The Quantum Entropy Cone of Hypergraphs, SciPost Phys. 9 (2020) 5, 067 [2] F. Verstraete, J. Dehaene, B. De Moor, H. Verschelde, "Four qubits can be entangled in nine different ways", Phys. Rev. A, 65: 052112 (2002). [3] A. Higuchi, A. Sudbery "How entangled can two couples get?" Phys. Lett. A 273 (2000), 213-217 [4] L. E. Danielsen, Database of Entanglement in Graph States https://www.ii.uib.no/~larsed/entanglement/ [5] M. Hein, W. Dür, J. Eisert, R. Raussendorf, M. Van den Nest, H.-J. Briegel, Entanglement in Graph States and its Applications, arXiv:0602096

---

## Editorial Decision

awaiting_resubmission